# Comparative Clinical Evaluation of a Novel FluA/FluB/SARS-CoV-2 Multiplex LAMP and Commercial FluA/FluB/SARS-CoV-2/RSV RT-qPCR Assays

**DOI:** 10.3390/diagnostics13081432

**Published:** 2023-04-16

**Authors:** Hyunseul Jee, Seoyeon Park, Junmin Lee, Chae Seung Lim, Woong Sik Jang

**Affiliations:** 1BK21 Graduate Program, Department of Biomedical Sciences, College of Medicine, Korea University, Seoul 02841, Republic of Korea; jhs603@korea.ac.kr; 2Department of Laboratory Medicine, College of Medicine, Korea University Guro Hospital, Seoul 08308, Republic of Korea; pseoy@kumc.or.kr (S.P.); dlwnsals15@korea.ac.kr (J.L.); malarim@korea.ac.kr (C.S.L.); 3Emergency Medicine, College of Medicine, Korea University Guro Hospital, Seoul 08308, Republic of Korea

**Keywords:** SARS-CoV-2, influenza A, influenza B, loop-mediated isothermal amplification (LAMP), multiplex LAMP

## Abstract

Influenza and coronaviruses cause highly contagious respiratory diseases that cause millions of deaths worldwide. Public health measures implemented during the current coronavirus disease (COVID-19) pandemic have gradually reduced influenza circulation worldwide. As COVID-19 measures have relaxed, it is necessary to monitor and control seasonal influenza during this COVID-19 pandemic. In particular, the development of rapid and accurate diagnostic methods for influenza and COVID-19 is of paramount importance because both diseases have significant public health and economic impacts. To address this, we developed a multi-loop-mediated isothermal amplification (LAMP) kit capable of simultaneously detecting influenza A/B and SARS-CoV-2. The kit was optimized by testing various ratios of primer sets for influenza A/B (FluA/FluB) and SARS-CoV-2 and internal control (IC). The FluA/FluB/SARS-CoV-2 multiplex LAMP assay showed 100% specificity for uninfected clinical samples and sensitivities of 90.6%, 86.89%, and 98.96% for LAMP kits against influenza A, influenza B, and SARS-CoV-2 clinical samples, respectively. Finally, the attribute agreement analysis for clinical tests indicated substantial agreement between the multiplex FluA/FluB/SARS-CoV-2/IC LAMP and commercial Allplex^TM^ SARS-CoV-2/FluA/FluB/RSV assays.

## 1. Introduction

Influenza and coronaviruses are highly contagious respiratory diseases that cause millions of cases and deaths annually worldwide [1,2]. Influenza typically peaks during winter months and causes approximately 1 billion cases, 3–5 million severe cases, and up to 650,000 deaths [3]. However, public health measures and travel restrictions implemented during the COVID-19 pandemic have significantly curbed global influenza circulation [4]. The influenza B/Yamagata lineage has not been detected since April 2020, and other influenza viruses with considerably lower genetic diversity have circulated [5]. As the COVID-19 pandemic gradually weakens owing to various quarantine and vaccination measures, epidemic prevention, and control, global travel is gradually returning to pre-epidemic levels. Therefore, it is necessary to develop measures to monitor and control the spread of seasonal influenza during the COVID-19 pandemic.

In response to this challenge, various multi-diagnostic real-time reverse transcription polymerase chain reaction (RT-PCR) assays that can simultaneously detect multiple respiratory pathogens, including influenza and COVID-19, have been developed. This is particularly useful during the COVID-19 pandemic because it can simultaneously diagnose influenza and COVID-19, which induce similar symptoms. Bouassa et al. developed an RT-qPCR assay that could simultaneously diagnose influenza, COVID-19, and respiratory syncytial virus (RSV), and this kit exhibited 97.9, 89.5, 97.6, and 100% accuracy for influenza A, influenza B, SARS-CoV-2, and RSV, respectively [6]. Additionally, as another diagnostic method, various rapid antigen tests that are capable of detecting both influenza and COVID-19 antigens in a single test are being developed [7]. Although RT-qPCR is a commonly used gold-standard method of diagnosis, it is relatively slow and requires specialized equipment and skilled personnel, limiting its usefulness to rapid diagnosis in resource-constrained settings [8]. Rapid diagnostic tests, which use antibodies to detect viral antigens or antibodies in patient samples, have the advantage of being quick but have low sensitivity and tend to have a high false-negative rate [9]. Therefore, an alternative diagnostic technique capable of accurately and quickly diagnosing viral infections is required.

Loop-mediated isothermal amplification (LAMP) is a molecular diagnostic technique that enables the rapid and sensitive detection of specific RNA or DNA sequences in target pathogens [10]. This technique amplifies the target nucleic acid sequence at a constant temperature, using a set of four to six primers that recognize distinct regions of the target sequence. Among these, two are loop primers that allow the reaction to occur more efficiently by promoting the formation of stem–loop structures in the amplification product [11]. The LAMP assay has several advantages over other nucleic acid amplification methods, such as being simple, sensitive, and rapid [12]. In addition, LAMP assays can be performed using a variety of detection methods, including colorimetric, lateral flow, fluorescence, and fluorescent probe detection [13,14,15,16]. Among these methods, fluorescent probe detection is particularly useful for the real-time monitoring of the LAMP reaction. Among the various fluorescent probe techniques, the LAMP-assimilating probe consists of a fluorescently-tagged probe and complementary quencher-tagged probe. The assimilating probe was designed to hybridize with the target sequence during the LAMP reaction and was incorporated into the amplification products by Bst DNA polymerase, resulting in an increase in the fluorescent signal [17]. As amplification proceeds, the fluorescence intensity of the probe increases and can be monitored in real-time using a fluorescent detector to determine the presence or absence of the target pathogen. The use of LAMP-assimilating probes enables a highly sensitive and specific simultaneous detection of multiple pathogens in a single reaction, making them a powerful tool for molecular diagnostics in various settings [18].

In this study, we developed a FluA/FluB/SARS-CoV-2 multiplex LAMP assay that can detect influenza A, influenza B, SARS-CoV-2, and an internal control (actin beta) using LAMP primers and assimilating probes that were previously reported [19,20]. Among the five different ratios of LAMP primer sets evaluated, the ratio of 1:0.5:1:0.2 of influenza A, influenza B, SARS-CoV-2, and the internal control primer set showed the best performance in LAMP tests against the clinical samples of influenza A, influenza B, and SARS-CoV-2. Finally, the performance of the FluA/FluB/SARS-CoV-2 multiplex LAMP assay was compared with that of the commercial Allplex^TM^ SARS-CoV-2/FluA/FluB/RSV assay (Seegene, Seoul, Korea) for clinical samples.

## 2. Materials and Methods

### 2.1. Clinical Samples and RNA Extraction

The Korea Disease Control and Prevention Agency (KDCA) measured the SARS-CoV-2 titers using the plaque-forming unit (PFU) test and provided 20 strains, including one wild-type sample and 19 mutant samples. Influenza A H1N1, H1N1pdm09, H3N2, and influenza B virus were cultured at the Department of Laboratory Medicine, the Korea University Guro Hospital, of which the viral titers were measured using the TCID50 method. For clinical sensitivity testing, we used clinical SARS-CoV-2 nasopharyngeal swab (NP) (*n* = 96), influenza A NP (*n* = 117), and influenza B NP (*n* = 61) samples collected from SARS-CoV-2-, influenza A-, and influenza B-infected patients (from February 2018 to July 2022) at Korea University Guro Hospital. All clinical samples were confirmed using the Allplex^TM^ SARS-CoV-2 assay (Seegene Inc., Seoul, South Korea) and Anyplex II RV16 Detection Kit (Seegene, Inc., Seoul, South Korea). To assess the specificity and cross-reactivity, 135 NP swab specimens were tested from individuals with (102) and without (33) viral respiratory infections. Respiratory viral infections, as confirmed via PCR using the Anyplex II RV16 detection kit, included nine coronaviruses (KHU1, NL63, and 229E), three RSV A, three RSV B, three adenoviruses (AdV), three parainfluenza virus (PIV) types 1–4, three human bocaviruses (HboV), three human enteroviruses (HEV), three human rhinoviruses (HRV), and three metapneumoviruses (MPV). Nucleic acids were extracted from all samples using Zentrix (Biozenthech, Seoul, Korea), according to the manufacturer’s instructions. Briefly, 200 µL of the sample was dispensed into a 96-well extraction plate and nucleic acid was extracted through the respiratory virus process program. The study was conducted in accordance with the guidelines of the Declaration of Helsinki and approved by the Institutional Review Board of Korea University Guro Hospital (approval number: 2021GR0547).

### 2.2. Primer Design

The influenza A, influenza B, SARS-CoV-2, and internal control RT-LAMP primer sets used in this study have been previously reported by our study group [19,20]. Primer sets for influenza A, influenza B, and SARS-CoV-2 were designed to target conserved regions of segment 7, the nucleoprotein, and the RdRP gene, respectively. The internal control primer set was designed within a conserved region of human actin beta mRNA (NM_001101.5:c.287-c.498) [19,20]. For the multiplex probe design, two types of additional synthetic oligonucleotide sequences were designed and added to the 5′ end of the LB or LF primer of each LAMP primer set. The 5′ end of the multiplex probe was tagged with a fluorescent marker. To quench the fluorophore multiplex probe, two types of complementary synthetic oligonucleotide sequences tagged with BHQ1 or BHQ2 at the 3′ end were used. All LAMP primers and probes were synthesized by Macrogen Inc. (Seoul, Republic of Korea; Table 1).

### 2.3. FluA/FluB/SARS-CoV-2 Multiplex LAMP Assay

The RT-LAMP assay was performed using ELPIS RT-LAMP 2X Master Mix (Elpis-Biotech, Daejeon, Republic of Korea). For the FluA/FluB/SARS-CoV-2 multiplex LAMP assay, the reaction mixture was prepared with 12.5 μL of 2X Master Mix, 1 μL of influenza A LAMP primer mix, 0.5 μL of influenza B LAMP primer mix, 1 μL of SARS-CoV-2 RdRP gene LAMP primer mix, 0.2 µL of internal control LAMP primer mix, 540 nM of quencher 1 solution, 30 nM of quencher 2 solution, and 3 μL of sample RNA (final reaction volume: 25 μL). The compositions of influenza A and influenza B LAMP primer mix were 1 μM of two outer primers (F3 and B3) and 32 μM of two inner primers (FIP and BIP), 4 μM of loop LF primer, 6 μM of loop LF probe primer, and 10 μM loop LB primer. The compositions of the SARS-CoV-2 RdRP gene primer mix were 4 μM of two outer primers (F3 and B3) and 32 μM of two inner primers (FIP and BIP), 10 μM of loop LF primer, 4 μM of loop LB probe primer, and 6 μM of loop LB probe primer. The compositions of actin beta gene (internal control) primer mix were 4 μM of two outer primers (F3 and B3) and 32 μM of two inner primers (FIP and BIP), 10 μM of loop LF primer, 4 μM of loop LB probe primer, and 6 μM of loop LB probe primer. The RT-LAMP assay was performed using a CFX 96 Touch Real-Time PCR Detection System (Bio-Rad Laboratories, Hercules, CA, USA) at 62 °C for 40 min. The FAM, Texas Red, Hex, and Cy5 fluorescence channels were used to detect influenza A, influenza B, SARS-CoV-2 RdRP, and internal control, respectively.

### 2.4. RT-PCR

The performance of the FluA/FluB/SARS-CoV-2 multiplex LAMP assay was compared with that of the Allplex^TM^ SARS-CoV-2/FluA/FluB/RSV assay (Seegene, Seoul, Republic of Korea) using the CFX96 Touch Real-Time PCR Detection System (Bio-Rad, California, USA). The Allplex^TM^ SARS-CoV-2/FluA/FluB/RSV assay was performed according to the manufacturer’s instructions. The PCR cycling conditions of the Allplex^TM^ SARS-CoV-2/FluA/FluB/RSV assay were as follows: reverse transcription at 50 °C for 20 min, inactivation at 95 °C for 15 min, 2 cycles of 95 °C for 10 s, 60 °C for 40 s, and 72 °C for 20 s followed by 41 cycles of 95 °C for 10 s, 60 °C for 15 s, and 72 °C for 10 s with fluorescence detection at 60 °C and 72 °C.

### 2.5. Limit of Detection (LOD) Tests of the FluA/FluB/SARS-CoV-2 Multiplex LAMP Assay

To determine the detection limit of the FluA/FluB/SARS-CoV-2 multiplex LAMP assay, extracted RNA from culture broth samples of SARS-CoV-2 (1 × 10^3^ PFU/mL), influenza A H1N1 (2 × 10^6^ TCID50/mL), influenza A H1N1pdm09 (2 × 10^6^ TCID50/mL), influenza A H3N2 (2 × 10^6^ TCID50/mL), and influenza B (7 × 10^6^ TCID50/mL) were respectively mixed with RNA isolated from non-infected NP clinical sample. Each sample was serially diluted 10-fold from the original sample to obtain six levels. All tests were repeated three times and determined as the minimum concentration in a 10-fold dilution series, at which three replicates were amplified.

### 2.6. Statistics

To calculate the diagnostic agreement between the FluA/FluB/SARS-CoV-2 multiplex LAMP assay and AllplexTM SARS-CoV-2/FluA/FluB/RSV assay, positive predictive value (PPV), negative predictive value (NPV), and kappa value, and kappa statistics were used. Statistical analysis was performed using SPSS 18 statistical package (SPSS 18 Inc., Chicago, IL, USA). The kappa index analyzed using crosstabulation was interpreted according to the interpretation suggested by Landis and Koch [21]. A kappa index of <0, 0–0.20, 0.21–0.40, 0.41–0.60, 0.61–0.80, and 0.81–1.00 indicates poor, slight agreement, fair, moderate, substantial, and almost perfect agreement between tests, respectively.

## 3. Results

### 3.1. Optimization of the FluA/FluB/SARS-CoV-2 Multiplex LAMP Assay Primer Set

In the multiplex LAMP assay, since there may be interference reactions between each primer set, an important factor in multiplex LAMP development is to obtain an optimal primer set ratio that can detect all target diseases by adjusting the ratio between each primer set. For optimization of the FluA/FluB/SARS-CoV-2 multiplex LAMP assay primer set, five different concentration ratios of the influenza A, influenza B, SARS-CoV-2, and internal control primer sets (No. 1: 1:1:1:0.2, No. 2: 1:0.5:1:0.2, No. 3: 1:0.5:0.5:0.2, No. 4: 0.5:0.5:1:0.2, and No. 5: 1:1:0.5:0.2, respectively) were tested with non-infected NP RNA samples spiked with SARS-CoV-2 (1 × 10^7^ PFU/mL), influenza A H1N1 (5 × 10^6^ TCID50/mL), and influenza B (1 × 10^6^ TCID50/mL) (Table 2, Figure 1). Ratios No. 1 and No. 3 showed fast Ct values for influenza B and A, respectively, but they did not detect influenza A and SARS-CoV-2, respectively. Ratio No. 4 did not detect influenza A, and ratio No. 5 showed all signal detections but slower Ct values in all four signals compared with ratio No. 2. Among five different concentration ratios of primer sets, ratio No. 2 of the FluA/FluB/SARS-CoV-2 multiplex LAMP assay primer set (a ratio of four LAMP primer set of 1:0.5:1:0.2) showed the best performance. Therefore, ratio No. 2 (1:0.5:1:0.2) for influenza A, influenza B, SARS-CoV-2, and the internal control LAMP primer set was determined to be the optimum ratio for the FluA/FluB/SARS-CoV-2 multiplex LAMP assay (Table 2, Figure 1). Figure 1B and Table 3 show the performance of the FluA/FluB/SARS-CoV-2 multiplex LAMP assay against influenza A H1N1, influenza A, H1N1pdm09, influenza A H3N2, influenza B, SARS-CoV-2, and normal NP RNA samples at the optimum conditions (ratio of the influenza A, influenza B, and internal control primer sets was 1:0.5:1:0.2 at 62 °C).

### 3.2. Detection Limit of the FluA/FluB/SARS-CoV-2 Multiplex LAMP Assay and Allplex™ SARS-CoV-2/FluA/FluB/RSV Assay

Using the detection limit test for influenza A H1N1, H1N1pdm09, H3N2, influenza B, and SARS-CoV-2, the analytical performance of the FluA/FluB/SARS-CoV-2 multiplex LAMP assay was evaluated and compared with that of the Allplex™ SARS-CoV-2/FluA/FluB/RSV assay (Table 4). Each RNA sample, diluted 10-fold in six levels, was used for the detection limit testing of the Allplex™ SARS-CoV-2/FluA/FluB/RSV and FluA/FluB/SARS-CoV-2 multiplex LAMP assays. The detection limits of the Allplex™ SARS-CoV-2/FluA/FluB/RSV and FluA/FluB/SARS-CoV-2 multiplex LAMP assays were 10^−1^ PFU/mL/7 × 10^1^ TCID50/mL and 10^0^ PFU/mL/7 × 10^2^ TCID50/mL against SARS-CoV-2 and influenza B RNA samples, respectively. For influenza A H1N1, H1N1pdm09, and H3N2 RNA samples, the detection limits of the Allplex™ SARS-CoV-2/FluA/FluB/RSV assay and FluA/FluB/SARS-CoV-2 multiplex LAMP assay were 2 × 10^2^/7 × 10^8^/2 × 10^2^ TCID50/mL and 2 × 10^4^/7 × 10^10^/2 × 10^4^ TCID50/mL, respectively. In all test samples, the FluA/FluB/SARS-CoV-2 multiplex LAMP assay showed a one- or two-step higher detection limit compared to the Allplex™ SARS-CoV-2/FluA/FluB/RSV assay.

### 3.3. Comparison of Performance between the FluA/FluB/SARS-CoV-2 Multiplex LAMP and Allplex™ SARS-CoV-2/FluA/FluB/RSV Assays Using Clinical Samples

To confirm the performance of the FluA/FluB/SARS-CoV-2 multiplex LAMP assay, the sensitivities and specificities of the assay were compared with those of the Allplex™ SARS-CoV-2/FluA/FluB/RSV assay (Table 5). Both assays showed 100% specificity for normal NP clinical samples (n = 102). For influenza A NP clinical samples (n = 117), the sensitivities of the FluA/FluB/SARS-CoV-2 multiplex LAMP and Allplex™ SARS-CoV-2/FluA/FluB/RSV assays were 90.60% and 94.87%, respectively. The specificities of the FluA/FluB/SARS-CoV-2 multiplex LAMP assay for influenza A NP clinical samples were 100%, whereas those of the Allplex™ SARS-CoV-2/FluA/FluB/RSV assay were 99.03%. The positive predictive value (PPV) and negative predictive value (NPV) of the two kits were 100/90.27 and 99.10/94.44, respectively. For influenza B NP clinical samples (n = 61), the sensitivities of the FluA/FluB/SARS-CoV-2 multiplex LAMP assay and Allplex™ SARS-CoV-2/FluA/FluB/RSV assay were 86.89% and 88.52%, respectively. The specificities of the FluA/FluB/SARS-CoV-2 multiplex LAMP assay for influenza B NP clinical samples were 100%, whereas those of the Allplex™ SARS-CoV-2/FluA/FluB/RSV assay were 99.03%. The PPV and NPV of the two kits were 100/92.72 and 98.18/93.58, respectively. For SARS-CoV-2 NP clinical samples (n = 96), the sensitivities of the FluA/FluB/SARS-CoV-2 multiplex LAMP and Allplex™ SARS-CoV-2/FluA/FluB/RSV assays were 98.96% and 97.92%, respectively. The specificities of both the FluA/FluB/SARS-CoV-2 multiplex LAMP and Allplex^TM^ SARS-CoV-2/FluA/FluB/RSV assays for SARS-CoV-2 NP clinical samples were 100%. The PPV and NPV of the two kits were 100/99.03 and 100/98.08, respectively. These results indicate that the FluA/FluB/SARS-CoV-2 multiplex LAMP assay possesses comparable sensitivity and specificity to the Allplex^TM^ SARS-CoV-2/FluA/FluB/RSV assay for detecting influenza A/B and SARS-CoV-2 NP in clinical samples.

Next, we calculated the agreement between the FluA/FluB/SARS-CoV-2 multiplex LAMP and Allplex™ SARS-CoV-2/FluA/FluB/RSV assays using SPSS 18 statistical package (SPSS 18 Inc., Chicago, IL, USA) (Table 6). The kappa values for influenza A, influenza B, and SARS-CoV-2 NP clinical samples indicated substantial agreement between the two assays for these viruses. A kappa value of 0.845 for influenza A suggests almost perfect agreement, whereas a kappa value of 0.791 for influenza B suggests substantial agreement. The highest kappa value of 0.970 for SARS-CoV-2 NP clinical samples indicated an almost perfect agreement between the two assays for detecting SARS-CoV-2. Overall, these results suggest that the FluA/FluB/SARS-CoV-2 multiplex LAMP and Allplex™ SARS-CoV-2/FluA/FluB/RSV assays have good agreement for the detection of influenza A, influenza B, and SARS-CoV-2 NP clinical samples.

### 3.4. Cross-Reactivity Test

To confirm the absence of cross-reactivity of the FluA/FluB/SARS-CoV-2 multiplex LAMP assay with other common respiratory viruses, 33 NP swabs from patients with known infections with nine coronaviruses (229E, NL63, and OC43), three HEV, three AdV, three PIV, three MPV, three HboV, three HRV, and six RSV A/B were tested using the FluA/FluB/SARS-CoV-2 multiplex LAMP assay (Table 7). The FluA/FluB/SARS-CoV-2 multiplex LAMP assay showed negative results for Flu A, Flu B, and SARS-CoV-2 channels against 33 other common respiratory viruses. The internal controls of the kit were positive for all but three RSV B samples. These results suggest that the FluA/FluB/SARS-CoV-2 multiplex LAMP assay did not cross-react with other infectious viruses.

### 3.5. SARS-CoV-2 Mutant Test

As the coronavirus continues to spread, new strains are emerging. To confirm that the developed FluA/FluB/SARS-CoV-2 multiplex LAMP assay can detect various SARS-CoV-2 mutant strains, one SARS-CoV-2 wild-type and 19 SARS-CoV-2 mutant samples distributed by the KDCA were used to evaluate the performance of the kit (Table 8). The developed FluA/FluB/SARS-CoV-2 multiplex LAMP assay indicated positive reactions to all SARS-CoV-2 wild-type and mutant viruses tested.

## 4. Discussion

Presently, RT-qPCR is considered the most sensitive and specific molecular diagnostic test for infectious diseases, according to reports [22,23]. When new molecular diagnostic kits are developed, their performance is often evaluated using RT-qPCR as the reference standard [24,25]. Currently, LAMP is the most extensively researched and developed technique among various isothermal amplification methods. However, the detection limit of LAMP is similar or one to two orders of magnitude higher than that of RT-qPCR. Additionally, its sensitivity and specificity for clinical samples are reported to be similar to or slightly lower than those of RT-qPCR, ranging from 0 to 10% lower [23,26,27]. Although LAMP amplification can detect the target gene within 5–10 min when the target gene concentration is high, it does not demonstrate comparable sensitivity to RT-qPCR when the target gene concentration is low.

Despite these drawbacks, isothermal amplification methods remain attractive because of their isothermal amplification, rapidity, and ability to use relatively inexpensive equipment [27,28]. In fact, RT-qPCR kits take approximately 2–3 h to complete, but RT-LAMP analysis can detect several viruses simultaneously within 40 min because it does not require a reverse transcription step. Additionally, although the sensitivity and specificity of LAMP isothermal amplification may be lower than those of qPCR, it has higher sensitivity and specificity than rapid kits that mainly use antigen or antibody tests in the field [29,30]. Therefore, various methods incorporating isothermal amplification technology that can be used in facilities without the necessary equipment, or in the field, are being developed.

In this study, we optimized the primer set and reaction conditions of the FluA/FluB/SARS-CoV-2 multiplex LAMP assay and evaluated its sensitivity and specificity compared to those of the commercially available Allplex^TM^ SARS-CoV-2/FluA/FluB/RSV assay. The sensitivities of the FluA/FluB/SARS-CoV-2 multiplex LAMP assay for influenza A, influenza B, and SARS-CoV-2 NP clinical samples were 90.6%, 86.89%, and 98.96%, respectively. The specificity of the FluA/FluB/SARS-CoV-2 multiplex LAMP assay against non-infected samples was 100%, which was excellent in terms of ruling out false positives. When compared to the commercial Allplex™ SARS-CoV-2/FluA/FluB/RSV assay (94.87%, 88.52%, and 97.92%), sensitivity levels of the FluA/FluB/SARS-CoV-2 multiplex LAMP assay were slightly lower but still quite accurate. In addition, the agreement between the FluA/FluB/SARS-CoV-2 multiplex LAMP assay and the Allplex^TM^ SARS-CoV-2/FluA/FluB/RSV assay revealed high kappa values for influenza A, influenza B, and SARS-CoV-2 NP clinical samples, indicating substantial agreement between the two assays. Moreover, the FluA/FluB/SARS-CoV-2 multiplex LAMP assay showed no cross-reactivity with the other 11 respiratory viruses, which is a significant advantage over some commercial kits that have shown cross-reactivity with other viruses. Moreover, our LAMP kit demonstrated the ability to detect 20 SARS-CoV-2 mutants, including highly transmissible Delta and Omicron variants, which are crucial for identifying and controlling emerging COVID-19 outbreaks. These findings suggest that the multiplex LAMP assay is reliable for detecting influenza A, influenza B, and SARS-CoV-2 and can be used as an alternative to the Allplex^TM^ SARS-CoV-2/FluA/FluB/RSV assay.

Generally, LAMP diagnostic kits are being developed as a middle ground between rapid antigen/antibody kits, which are available for field use but may have limited sensitivity, and the qPCR assays, which have high sensitivity but require complex equipment and trained personnel. In this study, a multiplex LAMP assay was optimized, which showed reliable and efficient diagnostic performance for detecting influenza A, influenza B, and SARS-CoV-2. The assay could diagnose both influenza and SARS-CoV-2 within an hour without requiring complex equipment. Additionally, it showed similar performance to commercial qPCR kits, making it a useful alternative to commercial qPCR kits that may not be suitable for use in the field.

In this study, we have demonstrated that developing a multiplex LAMP assay for detecting influenza and SARS-CoV-2 simultaneously could have some benefits. Our results suggest that the developed LAMP kit could provide a more comprehensive diagnosis of respiratory infections, especially during flu season when both viruses may be circulating. The ability to diagnose both influenza and SARS-CoV-2 using a single test could reduce the need for multiple tests and save time and resources. Additionally, the LAMP kit we developed could be used in resource-limited settings, where access to sophisticated equipment is limited, and could provide faster results compared to traditional PCR assays. This could be particularly useful in outbreak situations where quick and accurate diagnosis is crucial for effective control and management. Overall, our findings suggest that a multiplex LAMP assay has the potential to be a valuable tool for diagnosing respiratory infections, and could complement existing diagnostic methods in both clinical and public health settings.

## 5. Conclusions

In conclusion, the FluA/FluB/SARS-CoV-2 multiplex LAMP assay shows a similar level of sensitivity to the Allplex™ SARS-CoV-2/FluA/FluB/RSV assay and commercial RT-qPCR in clinical testing, although the FluA/FluB/SARS-CoV-2 multiplex LAMP assay has a higher detection limit than that of the commercial RT-qPCR kit. Furthermore, the FluA/FluB/SARS-CoV-2 multiplex LAMP assay can detect multiple strains of SARS-CoV-2, including Delta and Omicron, without cross-reactivity to other respiratory viruses and demonstrates excellent specificity for uninfected samples. Additionally, the high kappa values obtained in the concordance assays using the Allplex^TM^ SARS-CoV-2/FluA/FluB/RSV assay suggest that the FluA/FluB/SARS-CoV-2 multiplex LAMP assay can serve as an alternative to this commercial kit. 

## Figures and Tables

**Figure 1 diagnostics-13-01432-f001:**
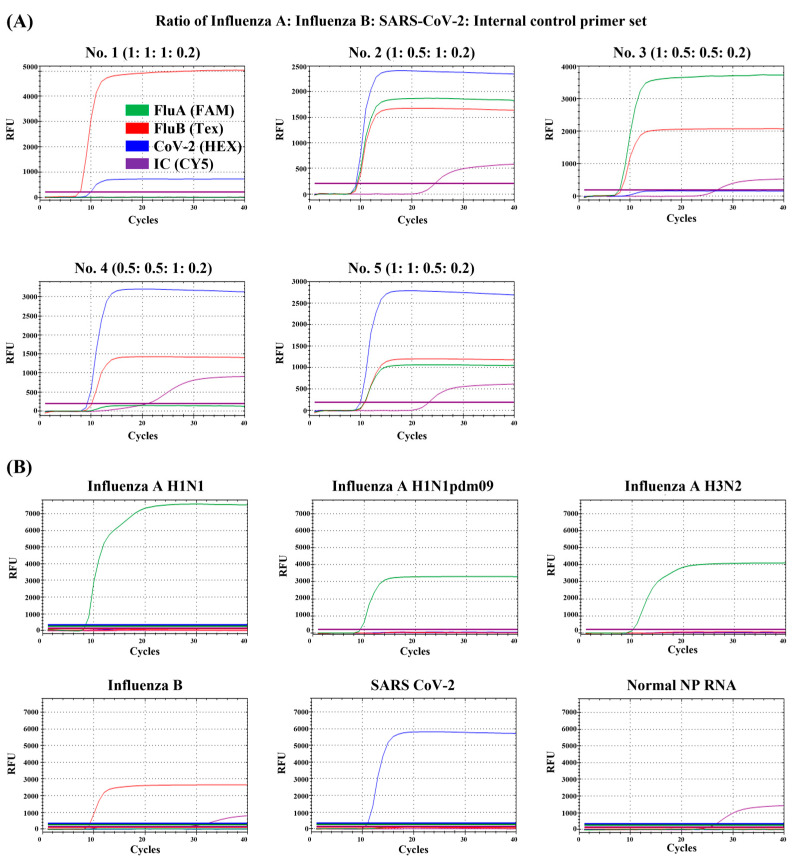
Optimization of the FluA/FluB/SARS-CoV-2 multiplex LAMP primer set. (**A**) Different concentration ratios of influenza A, influenza B, SARS-CoV-2, and internal control LAMP primer sets (No. 1–5) for cultured influenza A H1N1-, influenza B-, and SARS-CoV-2 (NCCP43346)-spiked NP RNA sample. (**B**) Performance of FluA/FluB/SARS-CoV-2 multiplex LAMP assay for influenza A H1N1, H1N1pdm09, H3N2, influenza B, SARS-CoV-2 (NCCP43346), and normal NP RNA.

**Table 1 diagnostics-13-01432-t001:** The FluA/FluB/SARS-CoV-2 multiplex LAMP assay primer sets used in this study.

Target	Name	Sequence (5′-3′)	µM
Influenza A	IAV F3	GAC TKG AAR RTG TCT TTG C	1
	IAV B3	TGT TRT TYG GGT CYC CAT T	1
	IAV FIP	TTA GTC AGA GGT GAC ARR ATT GCA GAT CTT GAG GCT CTC	32
	IAV BIP	TTG TKT TCA CGC TCA CCG TGT TTG GAC AAA GCG TCT ACG	32
	IAV BLP	CMA GTG AGC GAG GAC TG	10
	IAV FLP	GTC TTG TCT TTA GCC A	4
	IAV FLP probe 1	[FAM]-CGG GCC CGT ACA AAG GGA ACA CCC ACA CTC CGG TCT TGT CTT TAG CCA	6
Influenza B	IBV F3	GAG CTG CCT ATG AAG ACC	1
	IBV B3	CGT CTC CAC CTA CTT CGT	1
	IBV FIP	GAA CAT GGA AAC CCT TGC ATT TTA AGT TTT GTC TGC ATT AAC AGG C	32
	IBV BIP	GAA CAG RTR GAA GGA ATG GGR GCG ATC TGG TCA TTG GAG CC	32
	IBV BLP	TGC TGA TCT AGG CTT GAA TTC TGT	10
	IBV FLP	AGC TCT GAT GTC CAT CAA GCT CC	4
	IBV FLP probe 1	[TEX]-CGG GCC CGT ACA AAG GGA ACA CCC ACA CTC CGA GCT CTG ATG TCC ATC AAG CTC C	6
SARS-CoV-2 (RdRP gene)	RdRP F3	CCG ATA AGT ATG TCC GCA AT	4
	RdRP B3	GCT TCA GAC ATA AAA ACA TTG T	4
	RdRP FIP	ATG CGT AAA ACT CAT TCA CAA AGT CCA ACA CAG ACT TTA TGA GTG TC	32
	RdRP BIP	TGA TAC TCT CTG ACG ATG CTG TTT AAA GTT CTT TAT GCT AGC CAC	32
	RdRP LF	TGT GTC AAC ATC TCT ATT TCT ATA G	10
	RdRP LB	TCA ATA GCA CTT ATG CAT CTC AAG G	4
	RdRP LB probe 1	[HEX]-CGG GCC CGT ACA AAG GGA ACA CCC ACA CTC CGT CAA TAG CAC TTA TGC ATC TCA AGG	6
Human (Actin beta gene)	IC F3	AGT ACC CCA TCG AGC ACG	4
	IC B3	AGC CTG GAT AGC AAC GTA CA	4
	IB FIP	GAG CCA CAC GCA GCT CAT TGT ATC ACC AAC TGG GAC GAC A	32
	IC BIP	CTG AAC CCC AAG GCC AAC CGG CTG GGG TGT TGA AGG TC	32
	IC LF	TGT GGT GCC AGA TTT TCT CCA	10
	IC LB	CGA GAA GAT GAC CCA GAT CAT GT	6
	IC LB Probe 2	[CY5]-GTC AGT GCA GGC TCC CGT GTT AGG ACG AGG GTA GGC GAG AAG ATG ACC CAG ATC ATG T	4
Quencher probe 1		GAG TGT GGG TGT TCC CTT TGT ACG GGC CCG-BHQ1	
Quencher probe 2		CCT ACC CTC GTC CTA ACA CGG GAG CCT GCA CTG AC-BHQ2	

**Table 2 diagnostics-13-01432-t002:** The sensitivity of the FluA/FluB/SARS-CoV-2 multiplex LAMP assay according to different concentration ratios of the influenza A, influenza B, SARS-CoV-2, and internal control LAMP primer sets.

No	Ratio of Primer Set Mixture (FluA:FluB:SARS-CoV-2:IC)	FluA/FluB/SARS-CoV-2 Multiplex LAMP Assay Cycle Threshold Values (Ct Values)				
		Channels	FAM	Texas Red	HEX	Cy5
		Targets	Flu A H1N1	Flu B	SARS-CoV-2	IC-ACTB
1	1:1:1:0.2		N/A	7.69	9.90	N/A
2	1:0.5:1:0.2		9.27	9.44	9.08	24.46
3	1:0.5:0.5:0.2		7.92	8.21	N/A	27.03
4	0.5:0.5:1:0.2		N/A	10.12	9.23	23.93
5	1:1:0.5:0.2		10.76	10.92	10.01	23.39

N/A, not available.

**Table 3 diagnostics-13-01432-t003:** Performance of FluA/FluB/SARS-CoV-2 multiplex LAMP assay for influenza A (H1N1, H1N1pdm09, H3N2), influenza B, SARS-CoV-2, and NP RNA samples.

Sample	Concentration	FluA/FluB/SARS-CoV-2 Multiplex LAMP Assay Cycle Threshold Values (Ct Values)			
		FluA	FluB	RdRP	IC-ACTB
Influenza A H1N1	2 × 10^5^ TCID50/mL	8.28	N/A	N/A	N/A
Influenza A H1N1pdm09	7 × 10^12^ TCID50/mL	9.22	N/A	N/A	N/A
Influenza A H3N2	2 × 10^5^ TCID50/mL	10.17	N/A	N/A	N/A
Influenza B	7 × 10^5^ TCID50/mL	N/A	8.52	N/A	27.27
SARS-CoV-2	1 × 10^2^ PFU/mL	N/A	N/A	11.03	N/A
NP RNA sample		N/A	N/A	N/A	25.47

N/A, not available.

**Table 4 diagnostics-13-01432-t004:** Limit of detection test for Allplex™ SARS-CoV-2/FluA/FluB/RSV and FluA/FluB/SARS-CoV-2 multiplex LAMP assays for influenza A H1N1, H1N1pdm09, H3N2, influenza B, and SARS-CoV-2 RNA samples.

Virus	TCID50/mL	Allplex™ SARS-CoV-2/Flua/Flub/RSV Assay				FluA/FluB/SARS-CoV-2 Multiplex LAMP Assay			
		FluA	FluB	CoV-2	IC	FluA	FluB	CoV-2	IC
		Cycle Threshold Values (Ct Values)				Cycle Threshold Values (Ct Values)			
Influenza A H1N1	2 × 10^6^	18.83 ± 0.17	N/A	N/A	26.65 ± 0.24	8.55 ± 0.19	N/A	N/A	N/A
	2 × 10^5^	21.05 ± 0.08	N/A	N/A	30.99 ± 0.30	10.42 ± 0.19	N/A	N/A	N/A
	2 × 10^4^	24.01 ± 0.19	N/A	N/A	34.84 ± 0.04	13.15 ± 0.86	N/A	N/A	N/A
	2 × 10^3^	30.85 ± 0.51	N/A	N/A	N/A	N/A	N/A	N/A	N/A
	2 × 10^2^	37.32 ± 0.64	N/A	N/A	N/A	N/A	N/A	N/A	N/A
	2 × 10^1^	N/A	N/A	N/A	N/A	N/A	N/A	N/A	N/A
	2 × 10^0^	N/A	N/A	N/A	N/A	N/A	N/A	N/A	N/A
Influenza A H1N1pdm09	7 × 10^12^	18.05 ± 0.07	N/A	N/A	26.28 ± 0.07	8.95 ± 0.40	N/A	N/A	N/A
	7 × 10^11^	22.03 ± 0.30	N/A	N/A	29.94 ± 1.10	10.10 ± 0.61	N/A	N/A	N/A
	7 × 10^10^	25.81 ± 0.33	N/A	N/A	34.89 ± 1.34	13.4 ± 2.0	N/A	N/A	N/A
	7 × 10^9^	30.75 ± 0.51	N/A	N/A	N/A	N/A	N/A	N/A	N/A
	7 × 10^8^	36.29 ± 0.55	N/A	N/A	N/A	N/A	N/A	N/A	N/A
	7 × 10^7^	N/A	N/A	N/A	N/A	N/A	N/A	N/A	N/A
	7 × 10^6^	N/A	N/A	N/A	N/A	N/A	N/A	N/A	N/A
Influenza A H3N2	2 × 10^6^	18.11 ± 0.02	N/A	N/A	27.37 ± 0.29	9.29 ± 0.10	N/A	N/A	N/A
	2 × 10^5^	22.52 ± 0.10	N/A	N/A	32.31 ± 0.74	10.28 ± 0.67	N/A	N/A	N/A
	2 × 10^4^	26.93 ± 0.33	N/A	N/A	35.46 ± 0.65	11.92 ± 0.70	N/A	N/A	N/A
	2 × 10^3^	31.62 ± 0.48	N/A	N/A	N/A	N/A	N/A	N/A	N/A
	2 × 10^2^	35.19 ± 0.65	N/A	N/A	N/A	N/A	N/A	N/A	N/A
	2 × 10^1^	N/A	N/A	N/A	N/A	N/A	N/A	N/A	N/A
	2 × 10^0^	N/A	N/A	N/A	N/A	N/A	N/A	N/A	N/A
Influenza B	7 × 10^6^	N/A	17.24 ± 0.14	N/A	27.21 ± 0.40	N/A	7.21 ± 0.08	N/A	21.90 ± 1.94
	7 × 10^5^	N/A	19.36 ± 0.45	N/A	31.23 ± 1.08	N/A	8.12 ± 0.08	N/A	26.13 ± 0.30
	7 × 10^4^	N/A	23.28 ± 0.65	N/A	34.93 ± 0.87	N/A	9.48 ± 0.12	N/A	28.69 ± 0.46
	7 × 10^3^	N/A	27.11 ± 1.18	N/A	N/A	N/A	10.81 ± 0.18	N/A	35.37 ± 1.03
	7 × 10^2^	N/A	32.49 ± 0.70	N/A	N/A	N/A	13.27 ± 1.38	N/A	N/A
	7 × 10^1^	N/A	36.16 ± 1.08	N/A	N/A	N/A	N/A	N/A	N/A
	7 × 10^0^	N/A	N/A	N/A	N/A	N/A	N/A	N/A	N/A
SARS-CoV-2 (NCCP43346)	PFU/mL								
	1 × 10^0^	N/A	N/A	27.10 ± 0.08	30.86 ± 0.06	N/A	N/A	10.13 ± 0.11	N/A
	1 × 10^−1^	N/A	N/A	33.73 ± 0.67	34.62 ± 0.25	N/A	N/A	11.03 ± 0.04	N/A
	1 × 10^−2^	N/A	N/A	37.07 ± 1.03	N/A	N/A	N/A	N/A	N/A
	1 × 10^−3^	N/A	N/A	N/A	N/A	N/A	N/A	N/A	N/A
	1 × 10^−4^	N/A	N/A	N/A	N/A	N/A	N/A	N/A	N/A
	1 × 10^−5^	N/A	N/A	N/A	N/A	N/A	N/A	N/A	N/A
	1 × 10^−6^	N/A	N/A	N/A	N/A	N/A	N/A	N/A	N/A

N/A, not available.

**Table 5 diagnostics-13-01432-t005:** Comparison of performance of the FluA/FluB/SARS-CoV-2 multiplex LAMP assay and Allplex™ SARS-CoV-2/FluA/FluB/RSV assay for clinical samples.

Clinical Samples	Assay	P/N	Sensitivity (95% CI)	Specificity (95% CI)	PPV (95% CI)	NPV (95% CI)
Influenza A (n = 117)	Allplex™ SARS-CoV-2/FluA/FluB/RSV assay	111/6	94.87 (88.71–97.90)	99.03 (93.93–99.95)	99.10 (94.40–99.95)	94.44 (87.81–97.72)
	FluA/FluB/SARS-CoV-2 multiplex LAMP assay	106/11	90.60 (83.43–94-98)	100 (95.48–100)	100 (95.64–100)	90.27 (82.88–94.80)
Influenza B (n = 61)	Allplex™ SARS-CoV-2/FluA/FluB/RSV assay	54/7	88.52 (77.17–94.88)	99.03 (93.93–99.95)	98.18 (89.01–99.91)	93.58 (86.76–97.16)
	FluA/FluB/SARS-CoV-2 multiplex LAMP assay	53/8	86.89 (75.23–93.77)	100 (95.48–100)	100 (91.58–100)	92.72 (85.74–96.58)
SARS-CoV-2 (n = 96)	Allplex™ SARS-CoV-2/FluA/FluB/RSV assay	94/2	97.92 (91.96–99.94)	100 (95.48–100)	100 (95.12–100)	98.08 (92.55–99.97)
	FluA/FluB/SARS-CoV-2 multiplex LAMP assay	95/1	98.96 (93.51–99.94)	100 (95.48–100)	100 (95.16–100)	99.03 (93.93–99.95)

“P” and “N” indicate the positive and negative of the reaction, respectively. PPV, positive predictive value, NPV: negative predictive value. CI: confidence interval.

**Table 6 diagnostics-13-01432-t006:** Kappa values for calculating agreement between the FluA/FluB/SARS-CoV-2 multiplex LAMP assay and Allplex™ SARS-CoV-2/FluA/FluB/RSV assay.

		Allplex™ SARS-CoV-2/FluA/FluB/RSV Assay								
		Influenza A			Influenza B			SARS-CoV-2		
		P	N	Total	P	N	Total	P	N	Total
FluA/FluB/SARS-CoV-2 multiplex LAMP assay	P	100	6	106	46	7	53	93	2	95
	N	11	102	113	8	102	110	1	102	103
	Total	111	108	219	54	109	163	94	104	198
Cohen’s kappa index (*p*-value)		0.845 (<0.001)			0.791 (<0.001)			0.970 (<0.001)		
Strength of agreement		Almost perfect			Substantial			Almost perfect		

P, positive reaction; N, negative reaction.

**Table 7 diagnostics-13-01432-t007:** Cross-reactivity of the FluA/FluB/SARS-CoV-2 multiplex LAMP assay against other human infectious viruses.

Tested Clinical Samples	FluA/FluB/SARS-CoV-2 Multiplex LAMP Assay (Positive No./Test No.)			
	Flu A	Flu B	SARS-CoV-2	IC-ACTB
CoV 229E	0/3	0/3	0/3	3/0
CoV NL63	0/3	0/3	0/3	3/0
CoV OC43	0/3	0/3	0/3	3/0
HEV	0/3	0/3	0/3	3/0
AdV	0/3	0/3	0/3	3/0
PIV	0/3	0/3	0/3	3/0
MPV	0/3	0/3	0/3	3/0
HboV	0/3	0/3	0/3	3/0
HRV	0/3	0/3	0/3	3/0
RSV A	0/3	0/3	0/3	3/0
RSV B	0/3	0/3	0/3	0/3

**Table 8 diagnostics-13-01432-t008:** The performance of FluA/FluB/SARS-CoV-2 multiplex LAMP assay for one SARS-CoV-2 wild-type sample and 19 SARS-CoV-2 mutant samples.

Virus	Genotype (GSAID *)	Lineage	NCCP No.	FluA/FluB/SARS-CoV-2 Multiplex LAMP Assay			
				Flu A	Flu B	RdRP	IC-ACTB
				Cycle Threshold Values (Ct Values)			
SARS-CoV-2	GV (wild-type)	B.1.177	43346	N/A	N/A	7.12	N/A
	GRY	B.1.1.7	43381	N/A	N/A	7.29	N/A
	GH	B.1.351	43382	N/A	N/A	8.01	N/A
	GR	P.2	43383	N/A	N/A	7.03	N/A
	GH	B.1.427	43384	N/A	N/A	8.07	N/A
	GH	B.1.429	43385	N/A	N/A	8.35	N/A
	G	B.1.525	43386	N/A	N/A	12.78	N/A
	GH	B.1.526	43387	N/A	N/A	10.89	N/A
	GR	P.1	43388	N/A	N/A	7.12	N/A
	G	B.1.617.1	43389	N/A	N/A	8.06	N/A
	G	B.1.620	43404	N/A	N/A	7.33	N/A
	GK	B.1.617.2	43405	N/A	N/A	8.63	N/A
	GH	B.1.621	43407	N/A	N/A	8.35	N/A
	GRA	BA.1	43408	N/A	N/A	7.15	N/A
	GRA	BA.1.1	43411	N/A	N/A	8.20	N/A
	GRA	BA.2	43412	N/A	N/A	9.08	N/A
	GRA	BA.2.12.1	43423	N/A	N/A	8.19	N/A
	GRA	BA.2.3	43424	N/A	N/A	10.82	N/A
	GRA	BA.4	43425	N/A	N/A	9.75	N/A
	GRA	BA.5	43426	N/A	N/A	10.80	N/A

* GSAID: Global Initiative for Sharing All Influenza Data; N/A, not available.

## Data Availability

The authors declare that all related data are available from the corresponding author upon reasonable request.

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
