# Peer review of "Comparative Clinical Evaluation of a Novel FluA/FluB/SARS-CoV-2 Multiplex LAMP and Commercial FluA/FluB/SARS-CoV-2/RSV RT-qPCR Assays"

_diagnostics, 2023, doi:10.3390/diagnostics13081432_

Round 1

Reviewer 1 Report

In this work, Hyunseul et al. presented a Comparative Clinical Evaluation of a Novel FluA/FluB/SARS-CoV-2 Multiplex LAMP Assay and Commercial FluA/FluB/SARS-CoV-2/RSV RT-qPCR assay. The author has constructed a LAMP assay for the detection of multiplex virus and compared it with commercial FluA/FluB/SARS-CoV-2/RSV RT-qPCR assay. However, the author has not explained the outstanding advantages of his method compared to other methods and the necessity of this work. In addition, the data processing of the manuscript is not standardized, and the images are not clear. The author should carefully examine the manuscript.

1. The processing of manuscript data is not standardized, and there are many low-level errors. For example, the format of all sequences involved in the probe in Table 1 is incorrect, the data listed in Table 2 is not marked with the meaning of the data, the image in Figure 1 is not clear, and the table data has no standard deviation...... The author should carefully examine the manuscript.

2. The meaning of “N/A” should be indicated in the table.

3. The units in the manuscript should be unified. For example, TCID50/ml and PFU/ml.

4. Why the unit used for influenza is TCID50/ml and the unit used for SARS-CoV-2 is PFU/ml (for example, Table 3 and Table 4).

5. Compared to the commercial FluA/FluB/SARS-CoV-2/RSV RT-qPCR assay, the author's analytical method has no improvement in detection performance. What is the significance of this work? What are the advantages of this method compared to the author's previous multiple virus detection method? ( [1]. PLoS One 2020, 15, e0238615; [2]. PLoS One 2021, 16, e0248042 )

Reviewer 2 Report

Dear authors, 

I appreciate the comprehensiveness and technicality of the manuscript. Overall, I'm satisfied with the content and data presentation. A minor amendment can be considered for Table 2's caption in which you should also indicate the selected FAM, Texas Red, Hex, and Cy5 fluorescence channels for the detection.

Reviewer 3 Report

Please clarify or justify the description, "almost perfect agreement," for Kappa values.

Please explain the prevalence of the infections underlying the calculations of PPV and NPV. That is, what is the basis for calculating PPV and NPV. [I could not find "prevalence" in the manuscript].

The word "clinical" appears over 30 times in the manuscript, including in the title. No description of patients is provided. Adults or children, or both?  Symptomatic or asymptomatic, or both? ["Symptoms" appears once on line 47.]

"Performance" also appears several times, and as "clinical performance" in Table 5 and Please describe the setting and lines 218 and 220. Please define the characteristics of the clinical group(s) of patients underlying the use of "clinical performance."

Thank you for the opportunity of reviewing this paper.
